# Corrosion Resistance of Fe-Cr-Si Alloy Powders Prepared by Mechanical Alloying

Magdalena Sobota, Karolina Idczak, Robert Konieczny and Rafał Idczak *

Institute of Experimental Physics, University of Wrocław, pl. M. Borna 9, 50-204 Wrocław, Poland;
magdalena.sobota@uwr.edu.pl (M.S.); karolina.idczak@uwr.edu.pl (K.I.); robert.konieczny@uwr.edu.pl (R.K.)
* Correspondence: rafal.idczak@uwr.edu.pl

**Abstract:** Powders with nanometric crystallites of two ternary alloys $Fe_{0.90}Cr_{0.05}Si_{0.05}$ and $Fe_{0.85}Cr_{0.10}Si_{0.05}$ were prepared by mechanical alloying (MA) in a planetary high-energy ball mill at various milling times followed by annealing in a vacuum at 900 K to induce an oxygen-induced surface segregation of Cr and Si atoms. The prepared powders were characterized by X-ray diffraction (XRD), X-ray photoelectron spectroscopy (XPS) and scanning electron microscopy (SEM). The obtained results show that all prepared powders crystallize in the body-centered cubic structure and are composed of micrometric particles, which are polycrystalline and consist of many nanometric crystallites. The mean size of the particles as well as the crystallites decreases progressively with milling time. In order to study the anti-corrosion properties of the obtained materials, the powders were exposed to atmospheric gases at 870 K. After each oxidation step, the formation of iron oxides was investigated using $^{57}Fe$ transmission Mössbauer spectroscopy (TMS). It was found that the powders of $Fe_{0.90}Cr_{0.05}Si_{0.05}$ and $Fe_{0.85}Cr_{0.10}Si_{0.05}$ obtained after 10 and 20 h of MA are extremely resistant to oxidation. This result can be connected with the fact that XPS measurements reveal a high concentration of Cr and Si atoms on the surface of powder particles.

**Keywords:** corrosion; transmission Mössbauer spectroscopy; X-ray photoelectron spectroscopy; mechanical alloying; surface segregation; Fe-Cr-Si alloys





## 1. Introduction

Iron alloys are widely used in modern industry as structural materials. Since these alloys are often exposed to atmospheric conditions, corrosion becomes inevitable, resulting in the degradation and weakening of materials over time. The annual cost of the failure of corrosion-affected systems, as well as providing maintenance and implementing corrosion prevention methods, was recently estimated by the World Corrosion Organization, and is close to 3% of the Gross Domestic Product (GDP) of industrialized countries [1]. This enormous cost of the corrosion of iron has prompted many efforts to devise ways of reducing or preventing it [2,3].

Chromium is a metallic element well known for its good corrosion properties. The element is protected in oxidizing conditions by the formation of a $Cr_2O_3$ passive film, which acts as a barrier between the metal and the environment. The good anti-corrosion behaviour of chromium is the reason metals are alloyed with chromium, thereby making corrosion-resistant alloys [4,5]. Moreover, as was shown recently [6–10], the addition of silicon to Fe-Cr alloys greatly enhances their anti-corrosion properties since in the case of Fe-Cr-Si alloys in oxidizing conditions, the passive film composed of $Cr_2O_3$ and $SiO_2$ oxides is formed. Despite their excellent corrosion resistance, Fe-Cr-Si alloys are also of great interest in the research of magnetics and being used as electromagnetic wave absorbers for mobile phones, local area networks and radars [11,12]. However, it should also be noted that high concentrations of chromium and silicon contents in iron may lead to a deterioration in the mechanical properties of the alloy. In particular, due to the high content of Si, the

brittle ordered phases B2 and D0$_3$ are easily formed [13–15], while in Fe-Cr alloys, the formation of $\sigma$- and Cr-rich bcc phases are observed for a Cr concentration $\geq$10 at.% [16,17]. Fortunately, even for relatively low concentrations of Cr and Si solutes, good corrosion-resistant properties of iron-based alloys can be achieved through the phenomenon of surface segregation. As was shown, in the case of Fe-Cr, Fe-Si and Fe-Cr-Si alloys, the chromium and silicon atoms segregate to the alloys' surface resulting in the excellent anti-corrosion properties of these materials [7,9,18]. Moreover, it was found that despite the occurrence of these solutes' segregation during the sample preparation, the thermal treatment in an ultra-high vacuum (UHV) enhances this process. The observed effect could be explained by assuming a process known in the literature as oxygen-induced surface segregation [19,20]. The initial presence of oxygen atoms at the alloy's surface (as a contaminant from the atmosphere) leads to additional surface segregation of Si and Cr atoms at temperatures above 700 K. As a result, the higher concentrations of Cr and Si solutes at the surface lead to an increase in the corrosion resistance of the Fe-Cr and Fe-Cr-Si alloys [21].

In this paper, powders with nanometric crystallites of two ternary alloys Fe$_{0.90}$Cr$_{0.05}$Si$_{0.05}$ and Fe$_{0.85}$Cr$_{0.10}$Si$_{0.05}$ were prepared by mechanical alloying (MA) in a planetary high-energy ball mill at various milling times followed by annealing in a vacuum at 900 K to induce an oxygen-induced surface segregation of Cr and Si atoms. The main objectives of this work are (1) to examine the influence of the mean crystalline size of prepared powders on their anti-corrosion properties; and (2) to confirm that the oxygen-induced surface segregation of Cr and Si atoms can greatly enhance the corrosion resistance of the studied powders. In order to obtain these objectives, the prepared powders were characterized by X-ray diffraction (XRD), X-ray photoelectron spectroscopy (XPS) and scanning electron microscopy (SEM). To study the anti-corrosion properties of the obtained materials, the powders were exposed to atmospheric gases at 870 K. After each oxidation step, the formation of iron oxides was investigated using $^{57}$Fe transmission Mössbauer spectroscopy (TMS). The obtained results were discussed with respect to recent literature reports concerning the influence of crystalline size on the anti-corrosion properties of iron-based alloys. In particular, the corrosion resistance of Fe-Cr alloys is reported to be superior in nanocrystalline form when compared with that in a microcrystalline state [22–24]. At the same time, the studies of Fe-Cr-Si alloys revealed that the reduction in the mean crystalline size of mechanically synthesized Fe-Cr-Si alloys drastically decreases their corrosion-resistance properties [25]. Moreover, the comparison of the obtained results in this work with our previous findings presented in works [9,21,25] may provide valuable information about the impact of oxygen-induced surface segregation on the anti-corrosion properties of Fe-Cr-Si alloys.

## 2. Experimental Details and Data Analysis

### 2.1. Preparation of Samples

The Fe$_{0.90}$Cr$_{0.05}$Si$_{0.05}$ and Fe$_{0.85}$Cr$_{0.10}$Si$_{0.05}$ powders were prepared by MA in a high-energy planetary ball mill (Fritsch Pulverisette 6). The initial ternary mixtures in Fe:Cr:Si atomic ratios of 85:10:5 and 90:5:5 were prepared of powdered Fe (99.98%, Sigma-Aldrich, chips powdered in planetary mill at 300 rpm for 1 h), Cr (99.995%, Sigma-Aldrich, chips powdered in planetary mill at 300 rpm for 0.5 h) and Si (99.95%, Sigma-Aldrich, pieces powdered in planetary mill at 300 rpm for 5 min). A total of 10 g of each mixture was put in the grinding bowl with 25 grinding balls, and the mass ratio of balls to powder was equal to 10:1. The bowl and balls were made of tempered stainless steel. The milling was performed under argon atmosphere. The milling speed was set to 430 rpm for 10 h (marked as 10 h MA, only in the case of Fe$_{0.85}$Cr$_{0.10}$Si$_{0.05}$), 20 h (marked as 20 h MA) and 50 h (marked as 50 h MA). After that, the obtained powders were heated in a vacuum with the pressure lower than $10^{-4}$ Pa at 900 K for 24 h, and then the powders were slowly cooled to room temperature.

*2.2. Measurements and Data Analysis*

The structure of the polycrystalline powders was investigated by X-ray diffraction (XRD) at room temperature (RT) with a Panalytical Empyrean diffractometer with Cu K$_\alpha$ radiation ($\lambda$ = 1.5406 and 1.5444 Å). The collected XRD patterns were analyzed by the Rietveld method using the FULLPROF software package [26]. The Scherrer equation was used to determine the mean crystallite size $L$ of the prepared polycrystalline samples.

The morphology of the prepared materials was studied by scanning electron microscopy (SEM) Tescan Vega3 LMU. The microscope operated at a 30 kV acceleration voltage in high-resolution mode.

The surface's chemical composition and the chemical state of the constituent elements in the surface layer of the Fe$_{0.90}$Cr$_{0.05}$Si$_{0.05}$ and Fe$_{0.85}$Cr$_{0.15}$Si$_{0.05}$ powders obtained after 20 h of MA were studied by X-ray photoelectron spectroscopy (XPS). The powders were pressed into copper pellets.

The XPS measurements were performed in the ultra-high vacuum chamber, equipped with hemispherical analyzer SPECS Phoibos 150 with standard Mg and Al K$\alpha$ X-ray sources (photons energy 1254.6 eV and 1486.7 eV, respectively). A hemispherical analyzer was operated in Constant Analyzer Energy (CAE) mode, in which the pass energy is held at a constant value. The XPS spectrometer was calibrated to yield the typical values of the Au 4f doublet for the clean Au sample. All scans were taken at a photoelectron take-off angle of 90° at RT under a pressure lower than $10^{-8}$ Pa. The obtained high-resolution spectra (energy step ~0.08 eV, dwell time around 100 ms, number of each region scans up to 6 times) were analyzed using the CasaXPS program: the background of the spectra was subtracted using a software based on the Shirley method, and the fitting method was the Gaussian–Lorentzian (GL(30%)) method. The deconvolution of selected spectra was carried out in accordance with the expected atomic bonds at the surface, limited with the full width at half maximum (FWHM) not exceeding 2.5 eV for each component. In the case of Fe 2p spectra analysis, the Gupta–Sen multiplet peaks (GS) fitting method was used [27]. For these types of samples, the typical sampling depth for which the XPS signal can be detected is equal to 3.95 nm. The surface atomic concentration of each element $c_i$ was calculated using the formula:

$$c_i = \frac{I_i/\sigma_i \cdot \lambda_i}{\sum_i I_i/\sigma_i \cdot \lambda_i} \cdot 100\%, \tag{1}$$

where $I$ is a selected XPS peak intensity, $\sigma$ is the Scofield parameter [28] and $\lambda$ is the inelastic mean free path of an electron with a certain kinetic energy related to the XPS core-level line [29]. The presented $c_i$ values have an uncertainty of less than 2%.

In order to study the anti-corrosion properties of Fe-Cr-Si alloys, the samples were exposed to air at an elevated temperature. A small amount of the studied powders was placed in a quartz crucible and heated in a tube furnace at 870 K for the selected period of time. During the heating period, the powders had direct contact with atmospheric gases at ambient pressure. After each oxidation step, the $^{57}$Fe transmission Mössbauer spectroscopy (TMS) spectrum at room temperature was taken. All spectra were measured in transmission geometry with a conventional constant-acceleration spectrometer, using a 3.7 GBq $^{57}$Co-in-Rh standard source with a full width at half maximum (FWHM) of 0.22 mm/s. Each measured TMS spectrum was analyzed using a least-squares fitting procedure in terms of a sum of a different number of six-line patterns (sextets) corresponding to various isomer shifts (*IS*), a quadrupole shift (*QS*) and hyperfine fields (*B*) at $^{57}$Fe nuclei related to different chemical states of $^{57}$Fe Mössbauer probes. The fitting procedure was conducted under the thin absorber approximation. For each sextet, the two-line area ratio $I_{16}/I_{34}$ was constant and equal to 3/1. The ratio $I_{25}/I_{34}$ as well as three line widths $\Gamma_{16}$, $\Gamma_{25}$ and $\Gamma_{34}$ were free parameters. All the *IS* values presented in this paper are related to the $\alpha$-Fe standard.

According to the literature, the TMS spectra of Fe-Cr-Si alloys with a body-centered cubic structure (bcc) can be described using an additive model [18,30–32]:

$$B(n_1^{Cr}, n_2^{Cr}, n_1^{Si}, n_2^{Si}) \;=\; B_0 + n_1^{Cr}\Delta B_1^{Cr} + n_2^{Cr}\Delta B_2^{Cr} + n_1^{Si}\Delta B_1^{Si} + n_2^{Si}\Delta B_2^{Si} \tag{2}$$

and

$$IS(n_1^{Cr}, n_2^{Cr}, n_1^{Si}, n_2^{Si}) \;=\; IS_0 + n_1^{Cr}\Delta IS_1^{Cr} + n_2^{Cr}\Delta IS_2^{Cr} + n_1^{Si}\Delta IS_1^{Si} + n_2^{Si}\Delta IS_2^{Si}, \tag{3}$$

where $n_i^{Cr}$ and $n_i^{Si}$ are the number of Cr or Si atoms, respectively, located in the *i*-th coordination shell of $^{57}$Fe nuclei, and the changes in *B* and *IS* caused by the presence of the Cr or Si atom in the *i*-th coordination shell of $^{57}$Fe nuclei are denoted by $\Delta B_i^{Cr}$, $\Delta IS_i^{Cr}$, $\Delta B_i^{Si}$, and $\Delta IS_i^{Si}$, respectively. The *QS* in the bcc structure is equal to 0. Finally, it was assumed that $\Delta B_1^{Cr} = -3.34$ T [33], $\Delta B_2^{Cr} = -2.25$ T [33], $\Delta B_1^{Si} = -2.47$ T [34], and $\Delta B_2^{Si} \approx 0$ T [34], and the following configurations $(n_1^{Cr}, n_2^{Cr}, n_1^{Si}) = (0,0,0)$, (1,0,0) , (0,1,0), (0,0,1), (2,0,0), (1,1,0), (0,1,1), (0,1,2) were taken into account. All additional components that appeared in the spectra were attributed to new Fe compounds that were formed by exposure of the samples to air at an elevated temperature.

## 3. Results and Discussion

### 3.1. Phase Formation and Crystal Structure

Figures 1 and 2 present the XRD patterns collected for the $Fe_{0.90}Cr_{0.05}Si_{0.05}$ and $Fe_{0.85}Cr_{0.10}Si_{0.05}$ samples, respectively. The analysis of XRD data reveals that the prepared powders crystallized in the body-centered cubic (bcc) structure. In the case of the $Fe_{0.90}Cr_{0.05}Si_{0.05}$ 50 h MA sample, a secondary $Cr_2O_3$ phase with a rhombohedral structure (space group R$\bar{3}$c) and lattice parameters $a = b = 0.5022(7)$ nm, and $c = 1.3702(3)$ nm was detected [35]. The contribution of the $Cr_2O_3$ phase estimated from the XRD pattern is relatively high and close to 27(1) wt.%. However, it should be noted that the validity of the estimation of the phase composition for this sample using the measured XRD pattern is questionable because of the raised background, which is a consequence of the Fe and Cr fluorescence from Cu radiation [36]. Nevertheless, this finding clearly shows that annealing at 900 K for 24 h in a vacuum with a pressure close to $10^{-4}$ Pa leads to the initial oxidation of Cr atoms in the $Fe_{0.90}Cr_{0.05}Si_{0.05}$ 50 h MA sample.

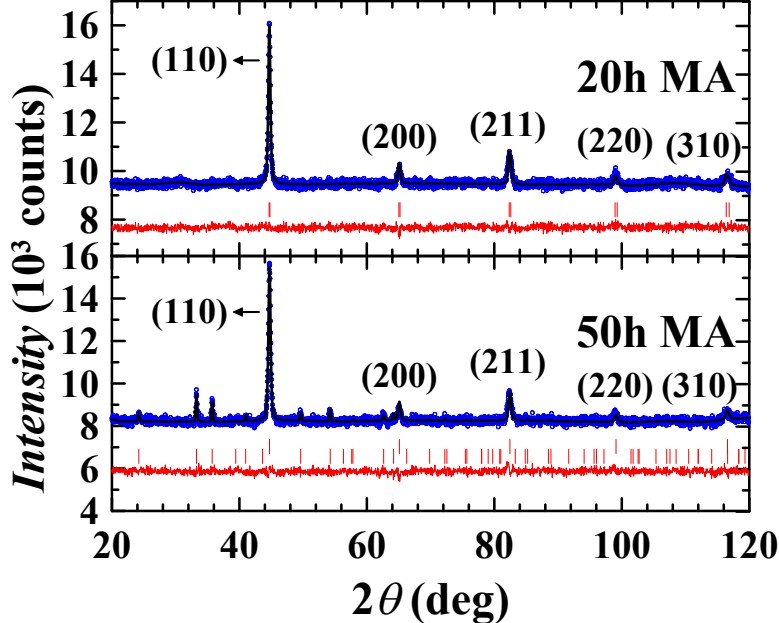

**Figure 1.** XRD patterns obtained for the $Fe_{0.90}Cr_{0.05}Si_{0.05}$ 20 h MA and 50 h MA samples annealed in vacuum at 900 K for 24 h. Blue dots and black lines represent the experimental data and the theoretical curves obtained from the Rietveld refinement, respectively. The red lines show the difference between the two. Red dashes indicate positions of the Bragg reflections (also described by their corresponding Miller indices).

The values of lattice constant *a* and mean crystallite size *L* calculated for the bcc phase are listed in Table 1. For all studied samples, the obtained *a* values are comparable with $a = 0.2867$ nm for pure α-Fe [37] as well as with *a* values determined for the mechanically alloyed $Fe_{0.85}Cr_{0.10}Si_{0.05}$ powders studied in work [25]. The lattice constants do not change significantly with the increment in milling time. At the same time, for both studied alloys, the mean crystallite size *L* gradually decreases with milling time.

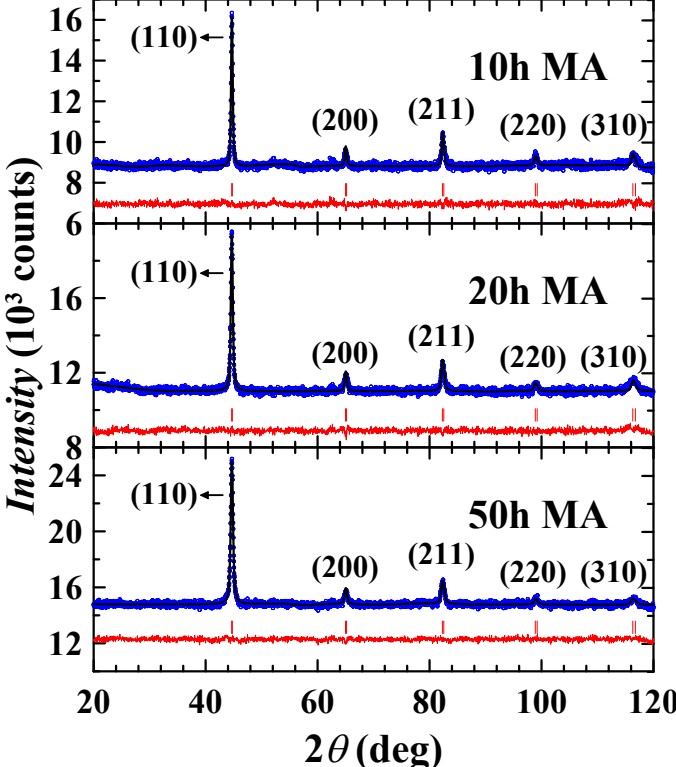

**Figure 2.** XRD patterns obtained for the $Fe_{0.85}Cr_{0.10}Si_{0.05}$ 10 h MA, 20 h MA and 50 h MA samples annealed in vacuum at 900 K for 24 h. Blue dots and black lines represent the experimental data and the theoretical curves obtained from the Rietveld refinement, respectively. The red lines show the difference between the two. Red dashes indicate positions of the Bragg reflections (also described by their corresponding Miller indices).

**Table 1.** The values of *a* and *L* parameters for bcc phase obtained from XRD patterns measured at RT for $Fe_{0.90}Cr_{0.05}Si_{0.05}$ and $Fe_{0.85}Cr_{0.10}Si_{0.05}$ samples.

| Sample | Space Group | *a* (nm) | *L* (nm) |
|---|---|---|---|
| | $Fe_{0.90}Cr_{0.05}Si_{0.05}$ | | |
| 20 h MA | Im$\bar{3}$m | 0.2866(1) | 36.2(2) |
| 50 h MA | Im$\bar{3}$m | 0.2863(1) | 28.6(1) |
| | $Fe_{0.85}Cr_{0.10}Si_{0.05}$ | | |
| 10 h MA | Im$\bar{3}$m | 0.2867(1) | 44.1(3) |
| 20 h MA | Im$\bar{3}$m | 0.2868(4) | 31.9(1) |
| 50 h MA | Im$\bar{3}$m | 0.2868(3) | 25.1(1) |

### 3.2. Morphology Analysis

Figures 3–7 present spherical and semi-spherical particles of the prepared $Fe_{0.90}Cr_{0.05}Si_{0.05}$ and $Fe_{0.85}Cr_{0.10}Si_{0.05}$ powders. The sizes of over 100 particles of each sample were measured and the results are presented in histograms. Assuming the log-normal distribution of the powders' particle sizes [38,39], the mean $\bar{D}$ and standard deviation $\sigma$ of the particles sizes were calculated and are listed in Table 2. As one can notice, the $\bar{D}$ and $\sigma$ parameters decrease

gradually with the milling time. Moreover, taking into account the mean crystallite sizes determined from XRD measurements, one can deduce that all of the prepared powders are composed of micrometric particles that are polycrystalline and consist of many nanometric crystallites. Comparing these results with an earlier study of $Fe_{0.85}Cr_{0.10}Si_{0.05}$ powders obtained by MA, it can be noticed that the shapes of the powder particles presented in work [25] are similar to those observed in this study. At the same time, the mean particle sizes reported in [25] are equal to 15.4 µm for 10 h MA, 9.0 µm for 20 h MA and 1.73 µm for 50 h MA. These significant differences can be explained by a different milling speed (500 rpm in work [25] instead of 430 rpm in this study) as well as by the fact that the previously obtained values of the mean particle sizes were determined for as-milled powders before heat treatment. In the case of the $Fe_{0.85}Cr_{0.10}Si_{0.05}$ samples annealed in a vacuum at 1270 K, the registered micrographs revealed that the particles were sintered together, and due to that, their exact particle size could not be determined [25]. Taking the above into account, it is clear that the annealing temperature of 900 K is too low to sinter $Fe_{0.90}Cr_{0.05}Si_{0.05}$ and $Fe_{0.85}Cr_{0.10}Si_{0.05}$ particles.

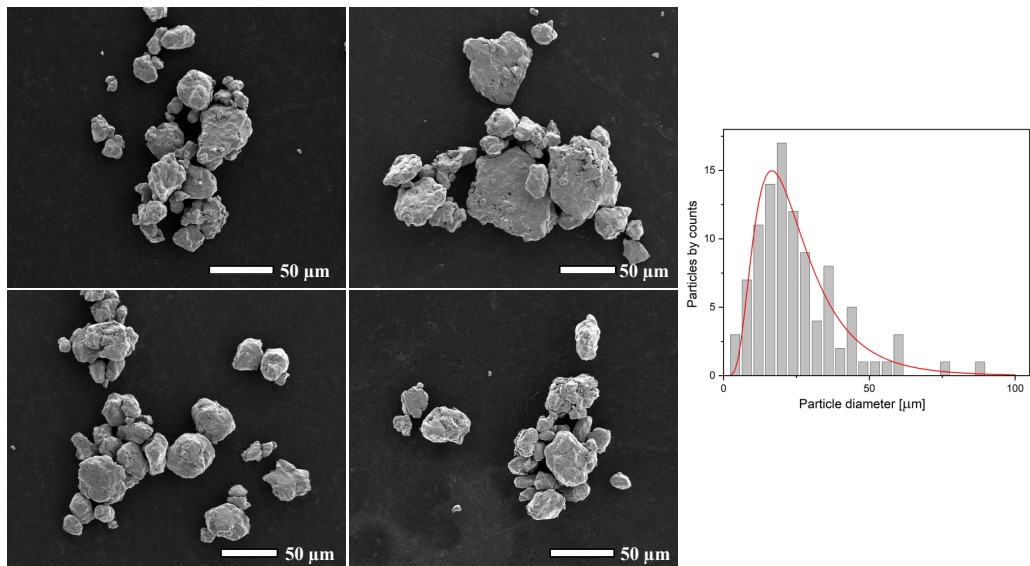

**Figure 3.** SEM micrographs (**left**) and sizes of about 100 particles (**right**) obtained for the $Fe_{0.90}Cr_{0.05}Si_{0.05}$ 20 h MA annealed in vacuum at 900 K for 24 h, fitted with log-normal distribution.

**Table 2.** The values of $\bar{D}$ and $\sigma$ parameters calculated using log-normal distribution of particle sizes of $Fe_{0.90}Cr_{0.05}Si_{0.05}$ and $Fe_{0.85}Cr_{0.10}Si_{0.05}$ powders.

| Sample | $\bar{D}$ (µm) | $\sigma$ (µm) |
|---|---|---|
| $Fe_{0.90}Cr_{0.05}Si_{0.05}$ | | |
| 20 h MA | 25.7(1.5) | 15.0(1.9) |
| 50 h MA | 10.6(0.7) | 8.1(1.1) |
| $Fe_{0.85}Cr_{0.10}Si_{0.05}$ | | |
| 10 h MA | 41.5(1.4) | 17.4(1.7) |
| 20 h MA | 26.4(1.1) | 14.8(1.4) |
| 50 h MA | 16.9(1.5) | 10.9(2.0) |

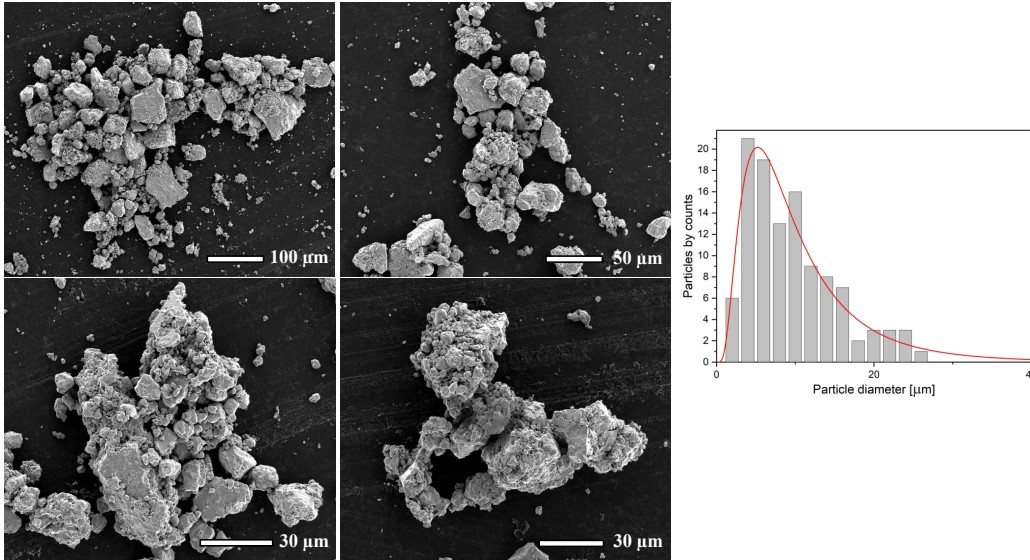

**Figure 4.** SEM micrographs (**left**) and sizes of about 100 particles (**right**) obtained for the $Fe_{0.90}Cr_{0.05}Si_{0.05}$ 50 h MA annealed in vacuum at 900 K for 24 h, fitted with log-normal distribution.

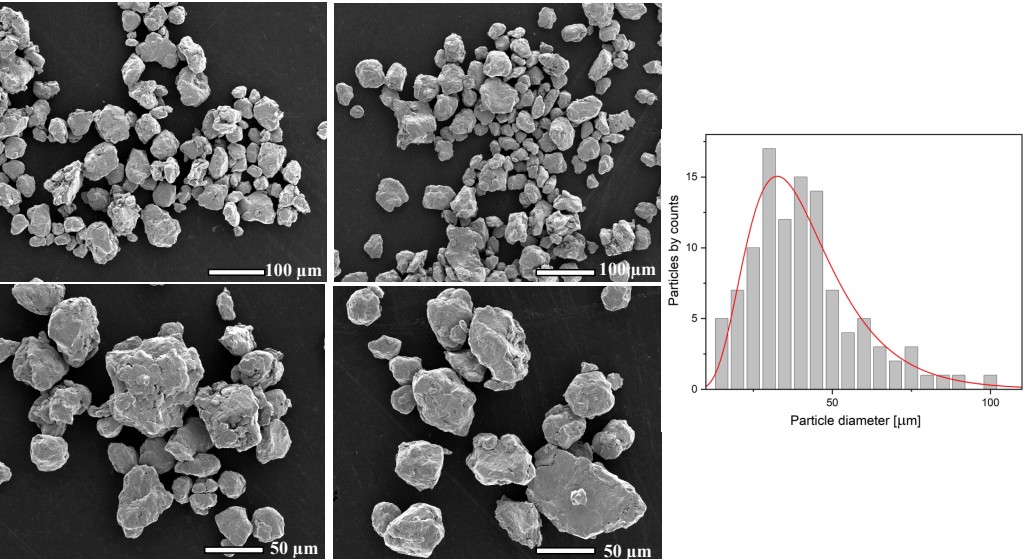

**Figure 5.** SEM micrographs (**left**) and sizes of about 100 particles (**right**) obtained for the $Fe_{0.85}Cr_{0.10}Si_{0.05}$ 10 h MA annealed in vacuum at 900 K for 24 h, fitted with log-normal distribution.

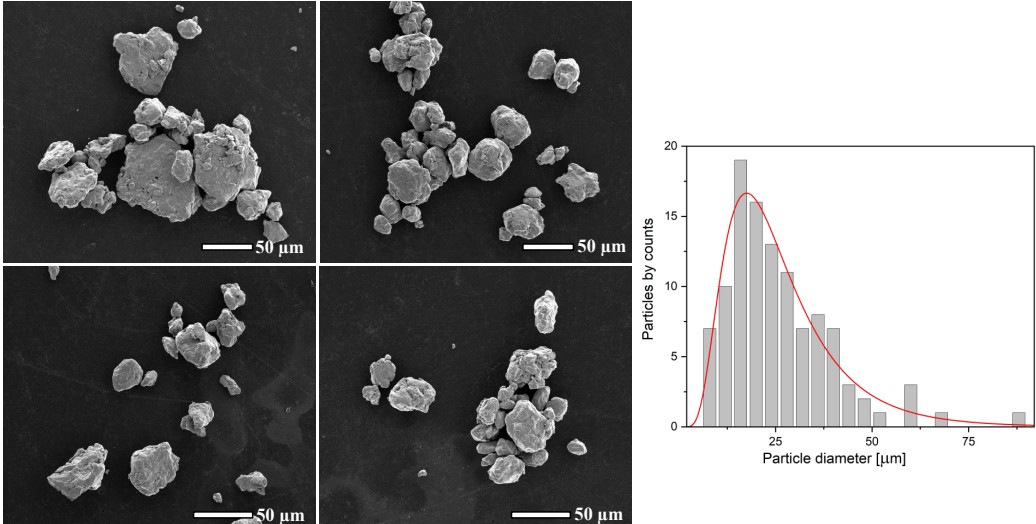

**Figure 6.** SEM micrographs (**left**) and sizes of about 100 particles (**right**) obtained for the $Fe_{0.85}Cr_{0.10}Si_{0.05}$ 20 h MA annealed in vacuum at 900 K for 24 h, fitted with log-normal distribution.

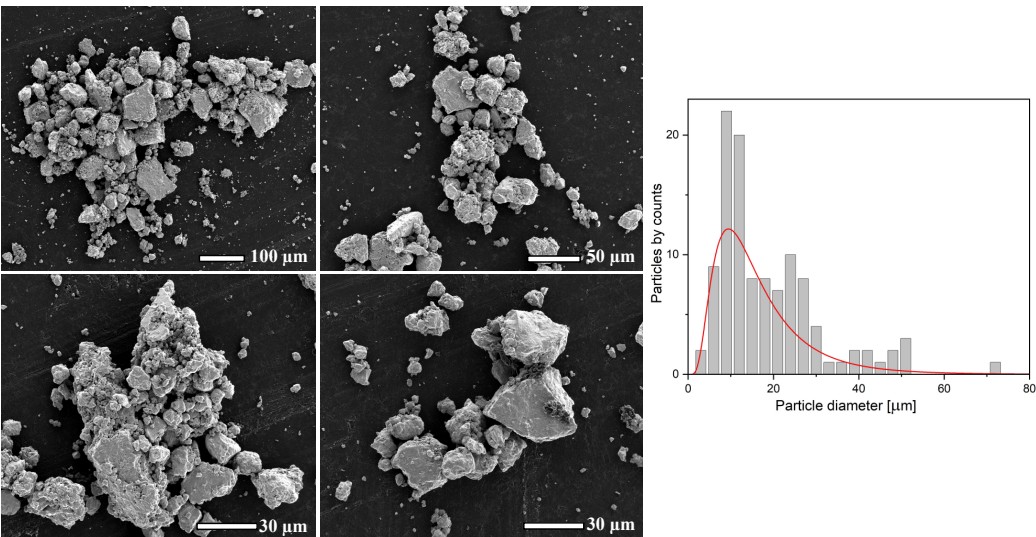

**Figure 7.** SEM micrographs (**left**) and sizes of about 100 particles (**right**) obtained for the $Fe_{0.85}Cr_{0.10}Si_{0.05}$ 50 h MA annealed in vacuum at 900 K for 24 h, fitted with log-normal distribution.

### 3.3. High-Temperature Corrosion Observed by Tms

TMS spectra recorded at room temperature for $Fe_{0.90}Cr_{0.05}Si_{0.05}$ 20 h MA and 50 h MA samples are shown in Figures 8 and 9, respectively. In the case of the $Fe_{0.90}Cr_{0.05}Si_{0.05}$ 20 h MA powder, which was annealed in a vacuum at 900 K for 24 h (marked as 'AP'), the measured TMS spectrum can be described by eight sextets related only to iron atoms located in the bcc Fe-Cr-Si alloy (see the additive model described in Section 2.2). The spectra collected for the powder after the oxidation process at 870 K for time $t \geq 32$ h reveal the presence of one additional sextet, which can be described by $IS = 0.375(7)$ mm/s, $QS = -0.19(1)$ mm/s and $B = 51.2(4)$ T. Based on the determined hyperfine parameters, this component can be ascribed to $\alpha$-$Fe_2O_3$ (hematite) [40,41]. In the case of $Fe_{0.90}Cr_{0.05}Si_{0.05}$ 50 h MA , the spectrum obtained for the AP sample is composed of only two sextets. The dominant one, with a relative intensity of 78% is related to Fe atoms without any Cr and Si atoms in their first two coordination shells (iron component). The second sextet described by $\Delta B = 24.7$ T corresponds to one Si atom located in the first coordination shell of the Fe atom

(Si component) [34]. This result indicates that the concentration of Cr in the bcc phase of the 50 h MA sample is much lower than for 20 h MA where the relative intensity of the iron component is close to 38% and the sextets related to Cr atoms in the nearest neighborhood of Fe are observed. The low concentration of Cr atoms in the bulk of $Fe_{0.90}Cr_{0.05}Si_{0.05}$ 50 h MA can be connected with the Cr segregation to the powder particle's surface [9]. Moreover, the mean crystallite size in this powder is equal to 28.6(1) nm. Therefore, it is plausible to assume that during thermal treatment most of the Cr solutes migrate into the particle's surface as well as into grain boundaries. This leads to the formation of powder particles that consist of a Cr-depleted core covered by a Cr-rich layer. Additionally, XRD data reveal the presence of large amounts of $Fe_2O_3$ in the 50 h MA sample. Taking into account that the grain boundaries in oxide scales have a strong effect on oxidation kinetics since they act as diffusion short circuits [42], it can be assumed that annealing at 900 K for 24 h in a vacuum with a pressure close to $10^{-4}$ Pa results in the oxidation of Cr atoms, which are mainly located in grain boundaries as well as on the particle's surface. All spectra recorded for the 50 h MA powder after the oxidation process consist of 4 sextets. Three of them are related to iron oxides. The sextet described by $IS = 0.374(1)$ mm/s, $QS = -0.18(1)$ mm/s and $B = 51.4(1)$ T corresponds to $\alpha$-$Fe_2O_3$, while two components with $IS = 0.273(9)$ mm/s, $QS \approx 0$ mm/s and $B = 48.9(1)$ T, and $IS = 0.611(2)$ mm/s, $QS \approx 0$ mm/s and $B = 45.6(3)$ T are ascribed to two different sites of Fe atoms in the $Fe_3O_4$ (magnetite) compound [40,43]. The last sextet is related to the iron component in the Fe-Cr-Si alloy. The absence of the Si component indicates that most of the silicon atoms dissolved in the bcc alloy were oxidized just after 1 h of the sample's exposure to air at 870 K.

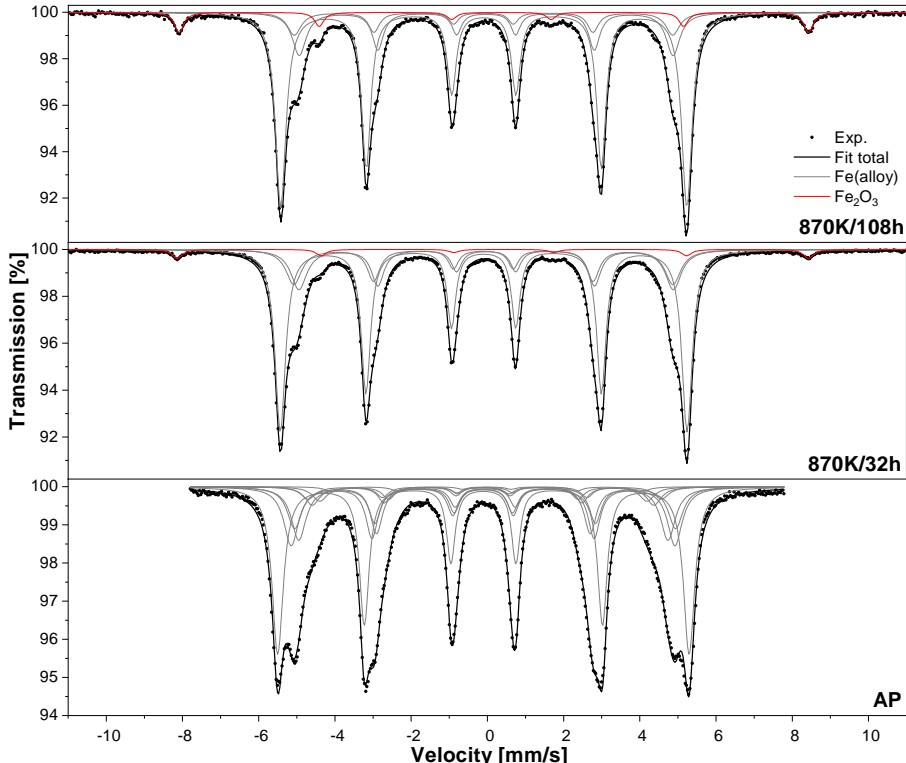

**Figure 8.** Selected TMS spectra measured at RT for the $Fe_{0.90}Cr_{0.05}Si_{0.05}$ 20 h MA sample.

In Figures 10 and 11, TMS spectra of $Fe_{0.85}Cr_{0.10}Si_{0.05}$ 10 h MA and 20 h MA are presented, respectively. In those spectra, no traces of iron oxides are found after 70 h of oxidation at 870 K, and the number of components ascribed to the bcc alloy does not decrease during oxidation. For the sample 50 h MA, TMS spectra are presented in Figure 12. In the case of the AP sample, the TMS spectrum is described only by eight sextets, which correspond to the $Fe_{0.85}Cr_{0.10}Si_{0.05}$ alloy. However, the relative intensities of components

that are related to the Si and Cr atoms located in the first two coordination shells of $^{57}$Fe (Si and Cr components) are much lower than those observed in the case of the 10 h MA and 20 h MA powders. Just as for the $Fe_{0.90}Cr_{0.05}Si_{0.05}$ 50 h MA sample, this finding indicates that the milling for 50 h and further annealing in a vacuum at 900 K for 24 h leads to the depletion of Cr and Si in the bcc phase of the Fe-Cr-Si alloy. After 1 h of the oxidation procedure, the TMS spectrum recorded for the $Fe_{0.85}Cr_{0.10}Si_{0.05}$ 50 h MA sample changes significantly. Similarly to the $Fe_{0.90}Cr_{0.05}Si_{0.05}$ 50 h MA powder, the Si and Cr components disappear. In addition, three sextets connected with the presence of $\alpha$-$Fe_2O_3$ and $Fe_3O_4$ are observed. Moreover, in the central part of the measured spectrum, the broad paramagnetic doublet with $IS = 0.36(1)$, $QS = 0.92(1)$ mm/s and the relative intensity of 5.7(3)% is detected. The origin of this component is uncertain. It could be connected with the presence of an amorphous $Fe_2O_3$ phase [44,45]. The TMS spectrum obtained for the sample 50 h MA after 20 h of oxidation reveals the presence of one sextet related to $\alpha$-$Fe_2O_3$.

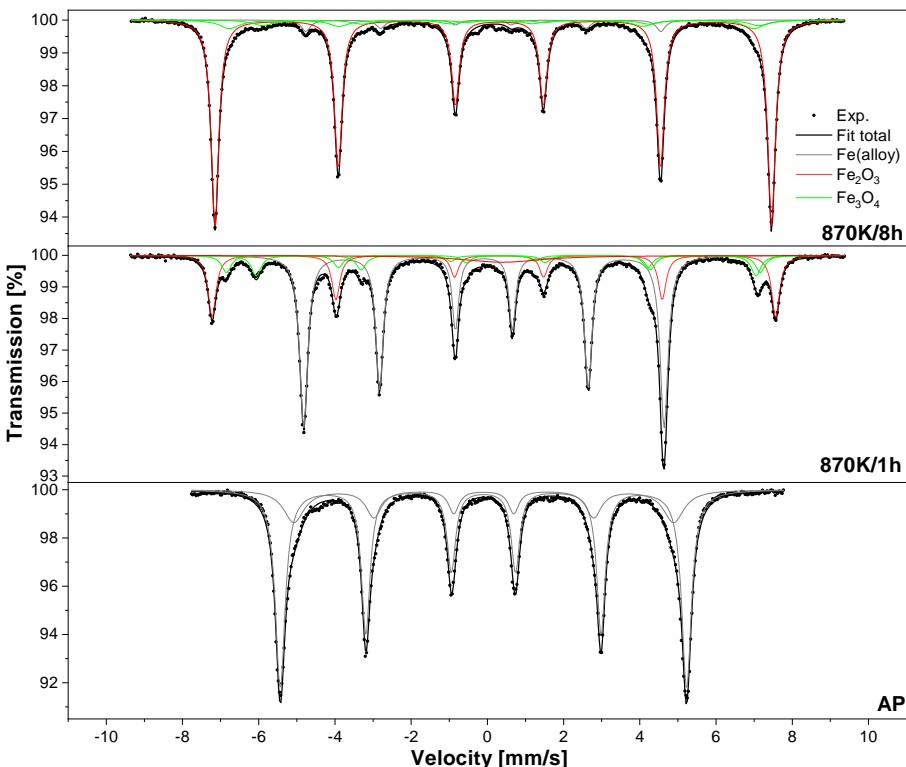

**Figure 9.** Selected TMS spectra measured at RT for the $Fe_{0.90}Cr_{0.05}Si_{0.05}$ 50 h MA sample.

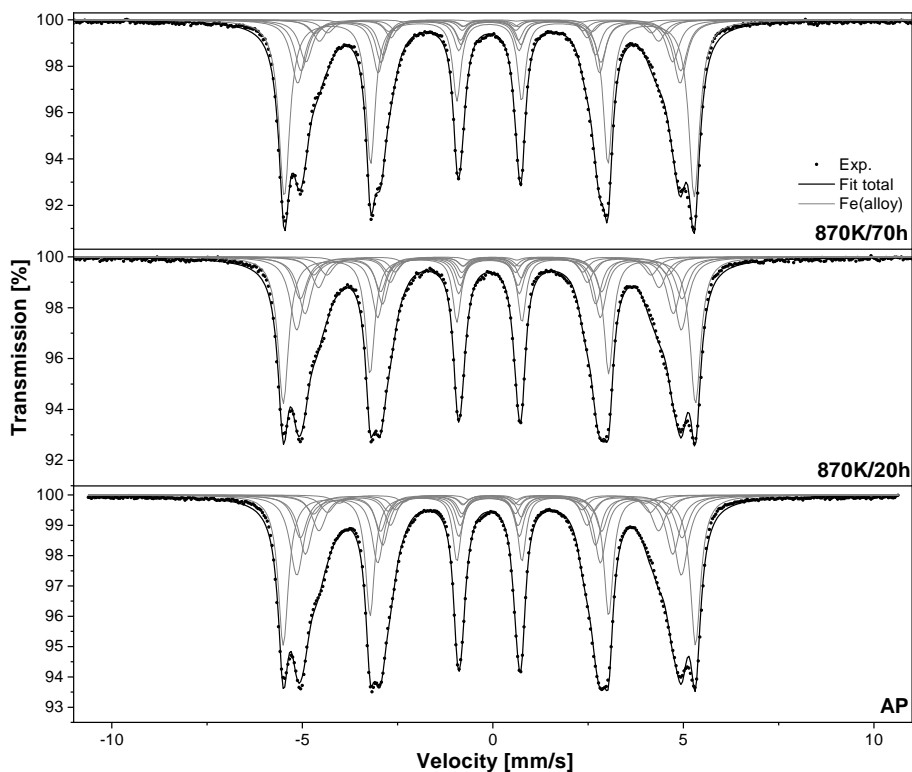

**Figure 10.** TMS spectra of the $Fe_{0.85}Cr_{0.10}Si_{0.05}$ 10 h MA annealed in vacuum at 900 K for 24 h.

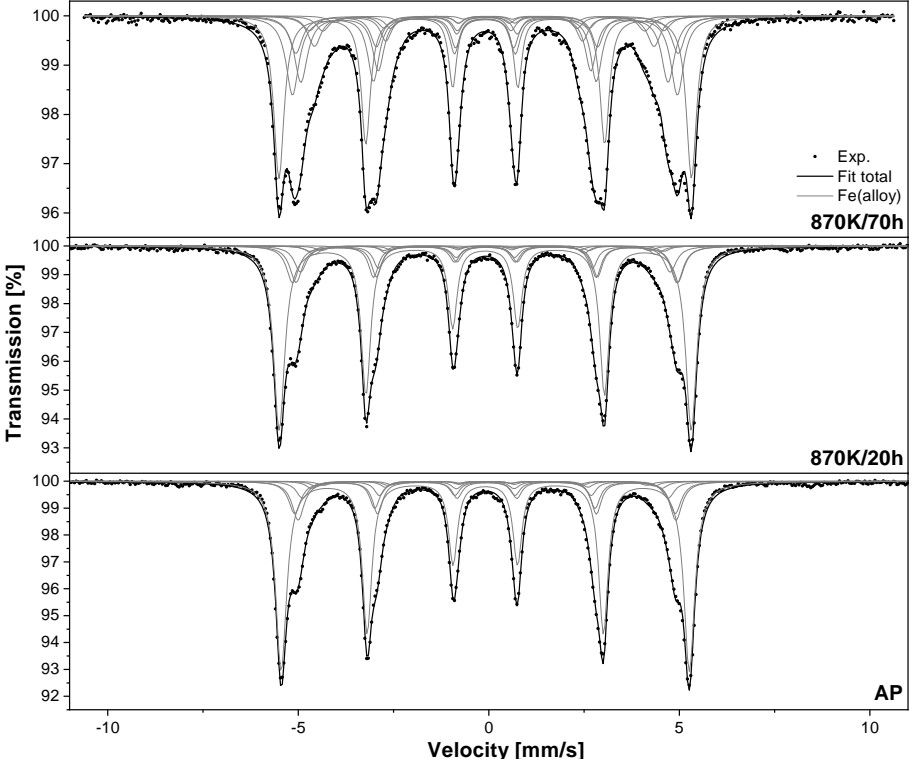

**Figure 11.** TMS spectra of the $Fe_{0.85}Cr_{0.10}Si_{0.05}$ 20 h MA annealed in vacuum at 900 K for 24 h. Spectra were recorded at RT.

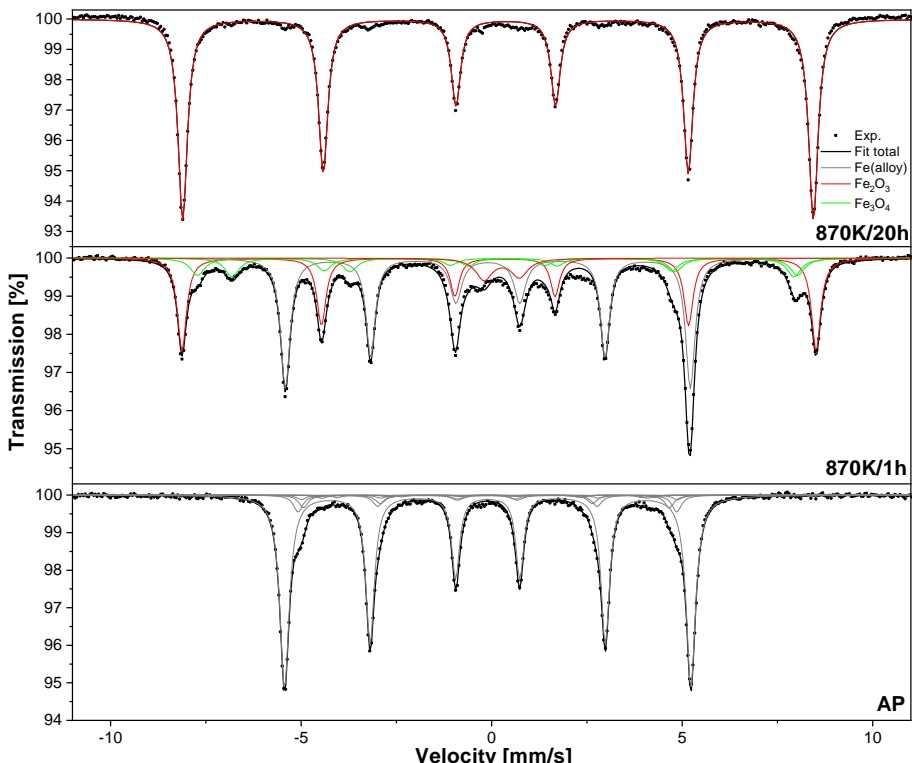

**Figure 12.** TMS spectra of the $Fe_{0.85}Cr_{0.10}Si_{0.05}$ 50 h MA annealed in vacuum at 900 K for 24 h. Spectra were recorded at RT.

The fractions of absorbing $^{57}Fe$ atoms in the bcc Fe-Cr-Si alloy ($C(alloy)$), $\alpha$-$Fe_2O_3$ ($C(Fe_2O_3)$) and $Fe_3O_4$ ($C(Fe_3O_4)$) oxides were calculated using the formula:

$$C(i) = \frac{\frac{I_i}{f_i}}{\sum_i \frac{I_i}{f_i}} \times 100\% \tag{4}$$

where $f$ is the Lamb–Mössbauer factor and $I$ is the relative intensity of the $i$-th component. In calculations, it was taken into account that the ratio of $f_{\alpha-Fe}$:$f_{\alpha-Fe_2O_3}$:$f_{Fe_3O_4}$ is equal to 1:1.08:1.05 [46]. The results are listed in Table 3. As one can notice, the powders prepared by MA for 10 h and 20 h are much more corrosion-resistant then those obtained after 50 h. This effect is mainly connected with the fact that the $L$ and $\bar{D}$ parameters decrease progressively with milling time leading to an increase in the specific surface area of the powder particles. It is worth noting here that these findings suggest that the influence of crystalline size on the anti-corrosion properties of the studied Fe-Cr-Si alloy is opposite to those observed for Fe-Cr alloys, for which nanocrystalline powders possess superior corrosion resistance when compared with those in a microcrystalline state [22–24]. Moreover, it should be mentioned that the oxidation process observed for both 50 h MA samples is strikingly faster than for 20 h MA. In particular, the 50 h MA samples that were exposed to air for 1 h contain more oxidized iron atoms than the 20 h MA ones after 108 h ($Fe_{0.90}Cr_{0.05}Si_{0.05}$) or 70 h ($Fe_{0.85}Cr_{0.10}Si_{0.05}$) of the oxidation procedure. At the same time, TMS spectra obtained for the 50 h MA samples show that these powders contain a Cr- and Si-depleted bcc phase, and this is probably another reason for the further deterioration in the anti-corrosion properties of these powders. The results obtained in this work can be compared with our previous corrosion studies on Fe-Cr-Si alloys. In work [25], the samples of $Fe_{0.85}Cr_{0.10}Si_{0.05}$ alloy obtained by the MA method were subjected to the same oxidation treatment. However, before the oxidation, they were annealed in a vacuum at 1270 K for 2 h. It was found that the previously studied $Fe_{0.85}Cr_{0.10}Si_{0.05}$ powders obtained after 10 h and 20 h of MA oxidize much faster than those studied in this work. In particular, for the 10 h MA and 20 h

MA samples after 70 h of oxidation treatment, $C(Fe_2O_3)$ levels are close to 3% and 56%, respectively [25]. This difference can be explained by taking into account the latest finding that the oxygen-induced surface segregation may enhance the corrosion resistance of Fe-Cr-Si alloys, and the optimal temperature at which this process has the highest efficiency is close to 900 K [21]. Therefore, it is clear that the $Fe_{0.85}Cr_{0.10}Si_{0.05}$ powders annealed at 900 K should have better anti-corrosion properties than those annealed at 1270 K. Another interesting finding is the fact that the $Fe_{0.85}Cr_{0.10}Si_{0.05}$ 10h and 20 h MA powders may have comparable corrosion resistance to $Fe_{0.85}Cr_{0.10}Si_{0.05}$ foils prepared by the traditional arc-melting technique [8]. In both cases, the TMS spectra recorded for the samples that were oxidized for an extremely long time do not reveal the presence of iron oxides. In the case of the $Fe_{0.90}Cr_{0.05}Si_{0.05}$ alloy, the foil annealed, before the oxidation treatment, at 1270 K for 2 h has worse anti-corrosion properties than the 20 h MA powder studied in this work since after 100 h of the oxidation procedure, the determined $C(Fe_2O_3)$ parameter is close to 22% [8], which is much higher than the 5.8% obtained in this study for the 20 h MA powder after 108 h of oxidation in similar conditions. This result confirms that the additional thermal treatment at 900 K greatly improves the corrosion resistance of Fe-Cr-Si alloys through the oxygen-induced surface segregation phenomenon [21]. Finally, it should be mentioned that the $Fe_{0.90}Cr_{0.05}Si_{0.05}$ alloy prepared by arc-melting and subsequently annealed at the optimal temperature of 900 K has slightly better anti-corrosion properties than the 20 h MA sample [21]. In fact, both materials were subjected to the same annealing and oxidizing conditions. The only difference was the larger specific surface area of the 20 h MA powder particles. Therefore, this difference is mainly responsible for the slightly faster corrosion observed in the 20 h MA sample.

**Table 3.** The fractions of absorbing $^{57}$Fe atoms in bcc Fe-Cr-Si alloy, $\alpha$-$Fe_2O_3$ and $Fe_3O_4$ oxides obtained from TMS spectra. The standard uncertainties for the $C$ parameters do not exceed 1%.

| Sample | | Exposure Time (h) | $C$(Alloy) (%) | $C(Fe_2O_3)$ (%) | $C(Fe_3O_4)$ (%) |
|---|---|---|---|---|---|
| $Fe_{0.90}Cr_{0.05}Si_{0.05}$ | 20 h MA | 32 | 97.0 | 3.0 | 0 |
| | | 80 | 95.6 | 4.4 | 0 |
| | | 108 | 93.8 | 6.2 | 0 |
| | 50 h MA | 1 | 57.3 | 26.5 | 16.2 |
| | | 2 | 20.3 | 59.6 | 20.1 |
| | | 6 | 5.1 | 81.4 | 13.5 |
| $Fe_{0.85}Cr_{0.10}Si_{0.05}$ | 10 h MA | 70 | 100 | 0 | 0 |
| | 20 h MA | 70 | 100 | 0 | 0 |
| | 50 h MA | 1 | 44.0 | 37.5 | 18.5 |
| | | 20 | 0 | 100 | 0 |

### 3.4. Surface Chemical Composition

In order to investigate the surface segregation process of Cr and Si atoms in $Fe_{0.90}Cr_{0.05}Si_{0.05}$ and $Fe_{0.85}Cr_{0.10}Si_{0.05}$ powders, the samples after 20 h of MA were studied by XPS. The measurements were performed for the as-prepared (marked as AP) powders as well as after the additional heating in UHV at 900 K for 15 min (marked as '900 K').

Figure 13 shows high-resolution XPS spectra obtained for the $Fe_{0.85}Cr_{0.10}Si_{0.05}$ alloy. The Si 2p spectra were recorded at the binding energy (BE) range of 85–100 eV. Unfortunately, in this BE region, the overlapping Fe 3s peaks are also observed. In the case of the AP sample, deconvolution allows designating four components at BE of 92.5, 96.0, 99.2 and 103.0 eV. The first two are assigned to Fe 3s core levels, the peak at 99.2 eV corresponds to metallic Si and the last one located at 103.0 eV is related to $Si^{4+}$ [9,21,47–49]. After heating at UHV, six components can be distinguished at: 91.0, 93.0, 95.3, 98.5, 101.8 and 103.6 eV. The first four components correspond to Fe 3s and the last two to $Si^0$ and $Si^{4+}$, respectively. The C 1s spectra were recorded at the 275–300 eV region and three components can be observed at: 285.3, 287.2 and 289.0 eV. In the case the annealed

sample, the number of components is the same and peaks are shifted to 284.3, 285.9 and 288.0 eV, respectively. These components can be ascribed to carbon contamination and carbon–oxygen bonds [21,50]. The O 1s spectra were measured at BE between 520 and 550 eV. For the AP sample, two components can be seen at 530.8 and 532.6 BE, and after heating, the peaks shift to 531.1 and 532.7 BE. These components correspond to water and carbon–oxygen bonds, respectively [9]. The spectra of the Cr 2p doublet were recorded at BE in the range of 570–600 eV. In the case of the AP sample, the doublet corresponding to metallic Cr is observed at 574.9 and 586.3 eV [21,51]. The second doublet located at 577.1 and 587.8 eV is ascribed to $Cr^{3+}$ species [21,52,53]. After heating at 900 K, three doublets can be observed: one for $Cr^0$ at 574.7 and 584.5 eV and two for $Cr^{3+}$ at 576.6 and 586.6 eV as well as at 578.5 and 588.5 eV. The Fe 2p spectra are measured at BE between 695 and 730 eV. In Figure 13e, only the Fe $2p_{3/2}$ envelope is presented. In the case of the AP sample, five components can be seen at 707.2, 709.5, 711.4, 713.3 and 715.2 eV. The first component corresponds to $Fe^0$ and the others to $Fe^{2+}$ and $Fe^{3+}$ species [21,54–56]. After heating, the $Fe^0$ can be observed at 706.8 BE and two components related to $Fe^{2+}$ and $Fe^{3+}$ can be seen at 708.1 and 709.7 eV, respectively.

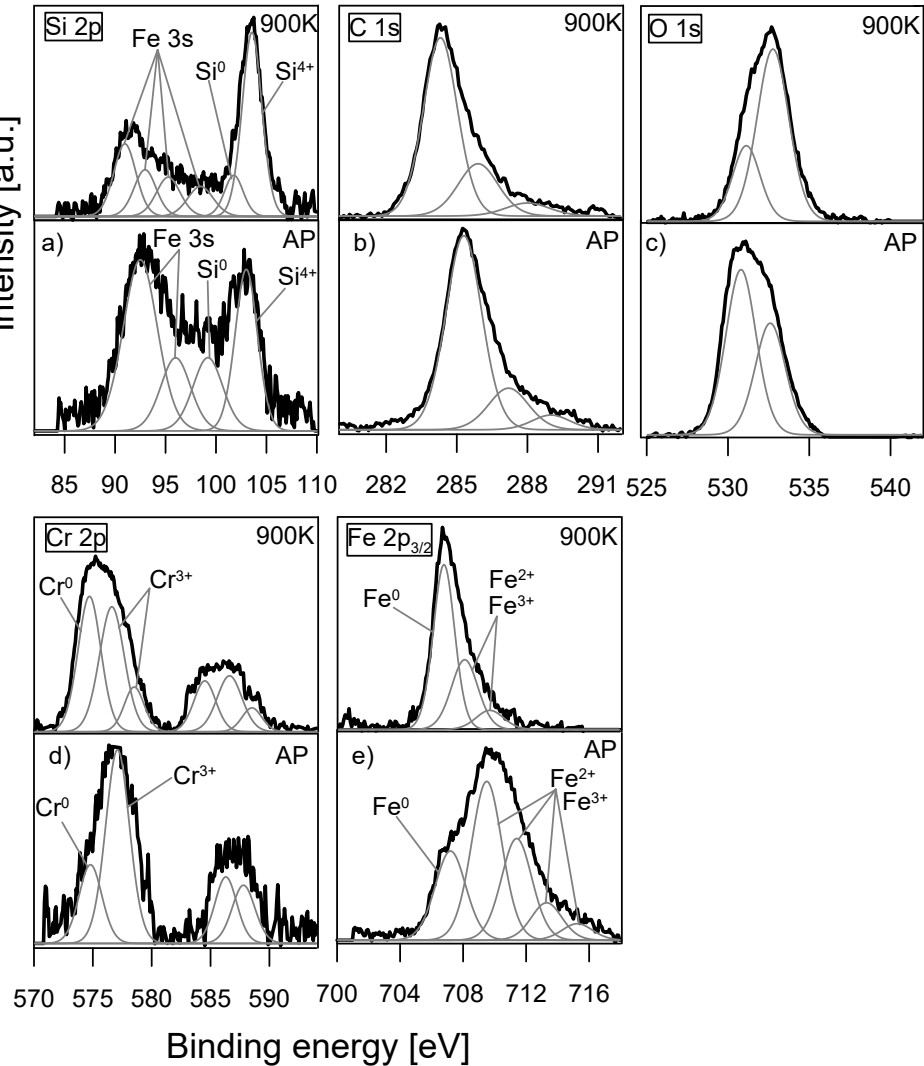

**Figure 13.** The selected XPS spectra: (**a**) Si 2p, (**b**) C 1s, (**c**) O 1s, (**d**) Cr 2p and (**e**) Fe $2p_{3/2}$ of the $Fe_{0.85}Cr_{0.10}Si_{0.05}$ 20 h MA annealed in vacuum at 900 K for 24 h marked as 'AP' and additionally annealed in UHV at 900 K/15 min marked as '900 K'.

The XPS spectra of $Fe_{0.90}Cr_{0.05}Si_{0.05}$ 20 h MA are presented in Figure 14. In the case of the Si 2p spectra measured for the AP sample, six components can be observed. Four

are related to Fe 3s core levels at 91.7, 94.0, 97.1 and 98.6 eV. The lines at BE of 100.6 eV and 103.4 eV represent $Si^0$ and $Si^{4+}$ species, respectively. After heating, three components of Fe 3s can be seen at 91.0, 93.4 and 96.0 eV, $Si^0$ is at 99.5 eV, and the position of the $Si^{4+}$ component has not changed. For the C 1s, three components can be observed for the AP sample at BE of 285.6, 287.2 and 289.1 eV, and for annealed at: 284.4, 286.1 and 288.0 eV. In the case of O 1s, for the AP sample, three peaks can be seen at: 530.6, 532.5 and 533.9 eV. For the sample heated in UHV, two components are visible: at 531.1 and 532.8 eV. The Cr 2p core levels for the AP sample reveal the presence of one doublet related to $Cr^0$ at 574.7 and 586.8 eV and two doublets ascribed to $Cr^{3+}$ species at: 576.7 and 588.4 eV, and 577.9 and 590.0 eV. After heating, the $Cr^0$ doublet is shifted to 574.6 and 584.3 eV, while the positions of the $Cr^{3+}$ doublets are changed slightly to: 576.4 and 586.3 eV, and 578.0 and 588.3 eV. In the case of the Fe $2p_{3/2}$ core levels, four components can be seen. The position of $Fe^0$ changes from 707.1 eV for the AP sample to 706.8 eV for the heated. At the same time, the peaks related to $Fe^{2+}$ and $Fe^{3+}$ species can be seen at 708.8, 710.5 and 712.5 eV (virgin sample) and at: 707.9, 709.3 and 711.4 eV (900 K sample).

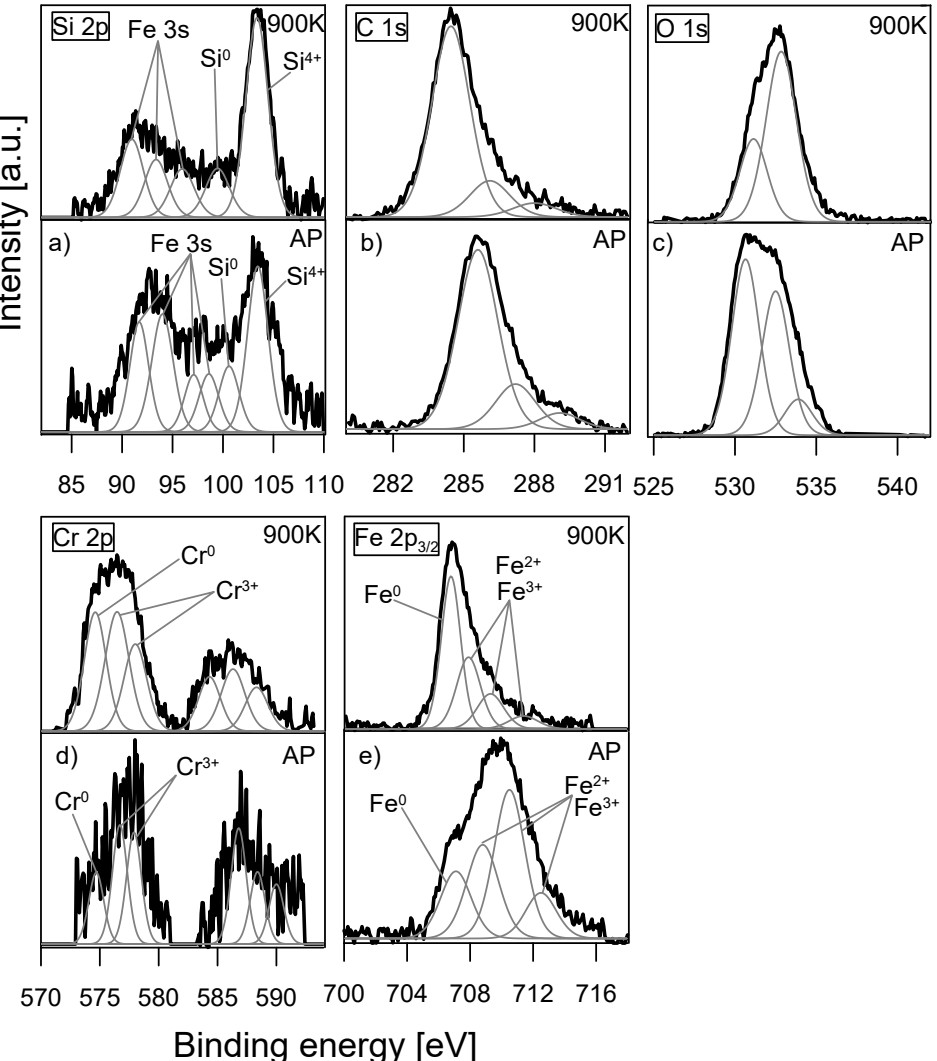

**Figure 14.** The selected XPS spectra: (**a**) Si 2p, (**b**) C 1s, (**c**) O 1s, (**d**) Cr 2p and (**e**) Fe $2p_{3/2}$ of the $Fe_{0.90}Cr_{0.05}Si_{0.05}$ 20 h MA annealed in vacuum at 900 K for 24 h marked as 'AP' and additionally annealed in UHV at 900 K/15 min marked as '900 K'.

The atomic surface concentrations $c_{Fe}$, $c_{Cr}$, $c_{Si}$ and $c_O$ are calculated using Equation (1). In the next step, the concentration ratios $c_{Cr}/c_{Fe}$, $c_{Si}/c_{Fe}$ and $c_O/c_{(Fe+Cr+Si)}$ are determined for both studied samples and compared with corresponding 'bulk' ratios that are calculated

with the assumption that the bulk chemical compositions of the synthesized alloys are equal to the nominal ones. All these data are presented in Table 4. As one can notice, for both AP samples, the $c_{Cr}/c_{Fe}$ ratios are more than 2 times higher than the bulk value, while the $c_{Si}/c_{Fe}$ ratios are more than 25 times higher than the value expected for the bulk samples. This result indicates that during the ball milling and further annealing at 900 K for 24 h, the surface segregation of Cr and Si atoms occurs and it is much more efficient in the case of Si atoms. The comparison between $c_{Cr}/c_{Fe} = 0.25$ and $c_{Si}/c_{Fe} = 2.00$ obtained in this work for the $Fe_{0.90}Cr_{0.05}Si_{0.05}$ 20 h MA powder and $c_{Cr}/c_{Fe} \approx 0$ and $c_{Si}/c_{Fe} = 1.04$ reported in work [21] for an arc-melted alloy, clearly shows that the segregation process of both solutes is greatly enhanced by annealing the powders at 900 K for 24 h. Moreover, additional heating in UHV induces a further increase in the $c_{Cr}/c_{Fe}$ and $c_{Si}/c_{Fe}$ ratios. Both these effects can be attributed to the oxygen-induced surface segregation process [21]. This assumption is based on the fact that the as-milled powders before annealing, as well as AP powders (after annealing at 900 K for 24 h), had direct contact with atmospheric gases, which leads to oxygen surface contamination. In the case of the AP samples, the determined ratio $c_O/c_{(Fe+Cr+Si)} > 1$ supports this interpretation. Finally, one can state that the XPS results confirm that the excellent anti-corrosion properties of both the 20 h MA samples are primarily a result of the high concentration of Cr and Si atoms on the surface of powder particles.

**Table 4.** Surface concentration ratios determined from XPS measurements for the alloys that were additionally thermally treated in UHV chamber.

| Sample | | | $c_{Cr}/c_{Fe}$ | $c_{Si}/c_{Fe}$ | $c_O/c_{(Fe+Cr+Si)}$ |
|---|---|---|---|---|---|
| $Fe_{0.90}Cr_{0.05}Si_{0.05}$ | surface | AP | 0.25 | 2.00 | 1.37 |
| | | 900 K | 1.48 | 3.69 | 0.77 |
| | bulk | | 0.06 | 0.06 | 0 |
| $Fe_{0.85}Cr_{0.10}Si_{0.05}$ | surface | AP | 0.27 | 1.59 | 1.14 |
| | | 900 K | 2.15 | 3.61 | 0.83 |
| | bulk | | 0.12 | 0.06 | 0 |

## 4. Conclusions

Powders with nanometric crystallites of two ternary alloys $Fe_{0.90}Cr_{0.05}Si_{0.05}$ and $Fe_{0.85}Cr_{0.10}Si_{0.05}$ were prepared by mechanical alloying in a planetary high-energy ball mill at various milling times followed by annealing in a vacuum at 900 K to induce an oxygen-induced surface segregation of Cr and Si atoms. The prepared powders were characterized by XRD, XPS, SEM and TMS. It was found that all prepared samples crystallize in the body-centered cubic structure with the lattice constant comparable to $\alpha$-Fe. SEM micrographs reveal that the powders are composed of micrometric particles and their mean size gradually decreases with the milling time.

In the second part of this work, the powders were exposed to air at 870 K for various periods of time and their anti-corrosion properties were characterized by TMS. The obtained results indicate that the powders of $Fe_{0.90}Cr_{0.05}Si_{0.05}$ and $Fe_{0.85}Cr_{0.10}Si_{0.05}$ obtained after 10 and 20 h of MA are extremely resistant to oxidation. This result can be connected with the fact that XPS measurements reveal a high concentration of Cr and Si atoms on the surface of powder particles, which is a result of the oxygen-induced surface segregation process. These findings will certainly contribute to the development of a new class of advanced materials for future applications in modern industry.

**Author Contributions:** Conceptualization, R.I. and M.S.; validation, R.I. and K.I.; formal analysis, R.I., M.S., R.K. and K.I.; investigation, M.S., R.K. and K.I.; resources, R.I. and K.I.; writing—original draft preparation, R.I. and M.S.; visualization, R.I. and M.S.; project administration, R.I.; funding acquisition, R.I. All authors have read and agreed to the published version of the manuscript.

**Funding:** This research received no external funding.

**Institutional Review Board Statement:** Not applicable.

**Informed Consent Statement:** Not applicable.

**Data Availability Statement:** The data presented in this study are available on request from the corresponding author.

**Conflicts of Interest:** The authors declare no conflict of interest. The funders had no role in the design of the study; in the collection, analyses, or interpretation of data; in the writing of the manuscript; or in the decision to publish the results.

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
