# Peer review of "Corrosion Resistance of Fe-Cr-Si Alloy Powders Prepared by Mechanical Alloying"

_coatings, doi:10.3390/coatings13101679_

Round 1

Reviewer 1 Report

In this work, the authors fabricated with mechanical alloying Fe-Cr-Si powders and assessed their microstructure and oxidation resistance. Please find my comment below:

1) Introduction needs to be expanded. Include more works on Fe-Cr powders that were fabricated by mechanical alloying. Discuss how the fabrication method affected the microstructure and attained properties.

2) What is the novelty of this work?

3) Expand the experimental part on the oxidation tests of the powder.

4) Assign peaks in XRD diffractograms (Figure 1 and 2)

5) The authors claim that the high concentration of Cr and Si atoms on the surface after oxidation indicated high resistance to corrosion. Nonetheless, it is known that stainless steels require at least 10.5 wt% Cr to form a protective film on the surface. Additionally, the low Cr containing stainless steels are not as protective as the grades containing larger amounts of Cr.

Since the oxidation testing of the powders is not clear, are you confident you exposed the powder to the needed conditions to properly assess their oxidation resistance?

Author Response

Letter of response

We thank very much the referee for carefully reading the paper and for numerous comments. In all cases they were taken into account.

Referee 1

In this work, the authors fabricated with mechanical alloying Fe-Cr-Si powders and assessed their microstructure and oxidation resistance. Please find my comment below:

1) Introduction needs to be expanded. Include more works on Fe-Cr powders that were fabricated by mechanical alloying. Discuss how the fabrication method affected the microstructure and attained properties.

Discussion: The Introduction has been improved in the revised manuscript.

2) What is the novelty of this work?

Discussion: The Introduction has been improved in the revised manuscript.

3) Expand the experimental part on the oxidation tests of the powder.

Discussion: In experimental part the details of the oxidation test have been added.

4) Assign peaks in XRD diffractograms (Figure 1 and 2)

Discussion: Figures 1 and 2 have been corrected.

5) The authors claim that the high concentration of Cr and Si atoms on the surface after oxidation indicated high resistance to corrosion. Nonetheless, it is known that stainless steels require at least 10.5 wt% Cr to form a protective film on the surface. Additionally, the low Cr containing stainless steels are not as protective as the grades containing larger amounts of Cr.

Since the oxidation testing of the powders is not clear, are you confident you exposed the powder to the needed conditions to properly assess their oxidation resistance?

Discussion: We agree that the low Cr containing stainless steels are not as protective as the grades containing larger amounts of Cr. However, in this work we studied Fe-Cr-Si alloys and as it was shown in works [6-10], the addition of Si to Fe-Cr system improves the anti-corrosion properties of this material. As it is mentioned in Introduction, in the case of Fe-Cr-Si alloys in oxidising conditions the passive film composed of Cr2O3 and SiO2 oxides is formed.

Reviewer 2 Report

Kindly find my minor revision comments:

1.     The title is clear and relevant to the content. However, consider rephrasing it slightly to make it more concise, for example: "Corrosion Resistance of Fe-Cr-Si Alloys Prepared by Mechanical Alloying."

2.     The abstract provides a good summary of the study, but it could be improved by mentioning the main findings or results. Include a sentence or two that highlights the key outcomes of the research.

3.     In the introduction, you discuss the importance of corrosion resistance in iron alloys but consider adding a brief sentence or two that explicitly states the objectives or goals of your study.

4.     In the last paragraph of the introduction, where you discuss the surface segregation of Cr and Si atoms, it would be helpful to briefly explain the significance of this phenomenon in the context of your research.

5.     In the section where you describe the XPS measurements (Section 2.2), consider providing a bit more detail about the instrumental setup and the specific conditions under which the measurements were performed. This can help readers understand the reliability of your XPS data.

6.     In the section where you discuss the morphology analysis (Section 3.2), consider including a brief discussion of how the particle size distribution may affect the corrosion resistance properties of the alloys. Link this information to the main findings of your study.

7.     When discussing the TMS spectra and the presence of iron oxides (Section 3.3), you could mention whether the observed iron oxide phases have any implications for the corrosion resistance of the materials. Do they enhance or hinder corrosion resistance? Connect this back to the main research question.

8.     Ensure that all figures and tables are properly referenced in the text.

9.     In Figure 2, consider adding labels to the x-axis and y-axis for clarity.

10.  In Figures 8, 9, 10, and 11, it would be helpful to include a legend or labels to identify the different sextets or components in the TMS spectra.

11.  Ensure that all figures and tables have appropriate captions that explain what they represent.

12.  Review the language for clarity and precision. For instance, in the abstract, consider rephrasing "The mean size of the particles gradually decreases with the milling time" to "The mean particle size decreases progressively with milling time" for better readability.

13.  Check for any grammatical or typographical errors.

14.  Add more references to the manuscript such as https://doi.org/10.3390/app13020730; https://doi.org/10.1007/s11661-022-06729-8; https://doi.org/10.1016/j.jallcom.2015.02.198; 10.1088/1742-6596/2267/1/012079; https://doi.org/10.1007/s11661-020-05758-5

Overall, the paper is well-organized and informative. Addressing these minor revisions will help improve clarity and reader understanding of your research.

Minor revision required

Author Response

Letter of response

We thank very much the referee for carefully reading the paper and for numerous comments. In all cases they were taken into account.

Referee 2

Kindly find my minor revision comments:

  1. The title is clear and relevant to the content. However, consider rephrasing it slightly to make it more concise, for example: "Corrosion Resistance of Fe-Cr-Si Alloys Prepared by Mechanical Alloying."

Discussion: The title has been corrected.

  1. The abstract provides a good summary of the study, but it could be improved by mentioning the main findings or results. Include a sentence or two that highlights the key outcomes of the research.

Discussion: We believe that the two last sentences in the abstract highlights the key outcomes of the research.

It was found that the powders of Fe0.90Cr0.05Si0.05 and Fe0.85Cr0.10Si0.05 obtained after 10 and 20 hours of MA are extremely resistant to oxidation. This result can be connected with the fact that XPS measurements reveal a high concentration of Cr and Si atoms at the surface of powder particles.

  1. In the introduction, you discuss the importance of corrosion resistance in iron alloys but consider adding a brief sentence or two that explicitly states the objectives or goals of your study.

Discussion: The Introduction has been improved in the revised manuscript.

  1. In the last paragraph of the introduction, where you discuss the surface segregation of Cr and Si atoms, it would be helpful to briefly explain the significance of this phenomenon in the context of your research.

Discussion: The Introduction has been improved in the revised manuscript.

  1. In the section where you describe the XPS measurements (Section 2.2), consider providing a bit more detail about the instrumental setup and the specific conditions under which the measurements were performed. This can help readers understand the reliability of your XPS data.

Discussion: We provide more details about the instrumental setup and the specific conditions under which the XPS measurements were performed.

  1. In the section where you discuss the morphology analysis (Section 3.2), consider including a brief discussion of how the particle size distribution may affect the corrosion resistance properties of the alloys. Link this information to the main findings of your study.

Discussion: In Section 3.3 we add the discussion about the influence of crystalline size on the anti-corrosion properties of the studied Fe-Cr-Si.

  1. When discussing the TMS spectra and the presence of iron oxides (Section 3.3), you could mention whether the observed iron oxide phases have any implications for the corrosion resistance of the materials. Do they enhance or hinder corrosion resistance? Connect this back to the main research question.

Discussion: In our opinion since the iron corrosion does not produce a reliable protective oxide film, the observed iron oxide phases do not have any positive influence on the corrosion resistance of the studied materials. In fact, the appearance of iron oxides decreases the corrosion resistance, the materials then become brittle and fragile, only segregated and oxidized Cr and Si atoms on the surface improve the anti-corrosion properties by forming a passivation film.

  1. Ensure that all figures and tables are properly referenced in the text.

Discussion: All figures and tables are properly referenced in the text.

  1. In Figure 2, consider adding labels to the x-axis and y-axis for clarity.

Discussion: In Fig. 2 there are labels of x-axis and y-axis.

  1. In Figures 8, 9, 10, and 11, it would be helpful to include a legend or labels to identify the different sextets or components in the TMS spectra.

Discussion: Figures 8, 9, 10, and 11 have been corrected.

  1. Ensure that all figures and tables have appropriate captions that explain what they represent.

Discussion: All figures and tables have appropriate captions that explain what they represent.

  1. Review the language for clarity and precision. For instance, in the abstract, consider rephrasing "The mean size of the particles gradually decreases with the milling time" to "The mean particle size decreases progressively with milling time" for better readability.

Discussion: Corrected.

  1. Check for any grammatical or typographical errors.

Discussion: The manuscript was checked for any grammatical or typographical errors.

  1. Add more references to the manuscript such as https://doi.org/10.3390/app13020730; https://doi.org/10.1007/s11661-022-06729-8; https://doi.org/10.1016/j.jallcom.2015.02.198; 10.1088/1742-6596/2267/1/012079; https://doi.org/10.1007/s11661-020-05758-5

Discussion: References https://doi.org/10.1007/s11661-022-06729-8 and https://doi.org/10.1007/s11661-020-05758-5 are already cited in the manuscript. References https://doi.org/10.3390/app13020730 and https://doi.org/10.1016/j.jallcom.2015.02.198 have been added.

Overall, the paper is well-organized and informative. Addressing these minor revisions will help improve clarity and reader understanding of your research.

Reviewer 3 Report

the manuscript is written well and the authors are requested to provide the mechanical property of the synthesized powder sample for more understanding to the readers. the manuscript can be accepted for publication after the inclusion of mechanical property of the material.

English correction is not required.

Author Response

Letter of response

We thank very much the referee for carefully reading the paper and for comments.

Referee 3

the manuscript is written well and the authors are requested to provide the mechanical property of the synthesized powder sample for more understanding to the readers. the manuscript can be accepted for publication after the inclusion of mechanical property of the material.

Discussion: In our opinion the study of mechanical properties of the synthesized powder samples are not directly connected with the main objectives of our work.

Reviewer 4 Report

In the present research, the authors try to fabrication the Fe-Cr-Si alloy powders by the mechanical alloy method. The paper exhibits some results but there are some questions.

1. The authors titled the paper as “Anti-Corrosion Properties of Fe-Cr-Si Alloys Prepared by Mechanical Alloying”. Based on the title, it seems the authors would fabricate the bulk Fe-Cr-Si alloy. Actually, they mainly prepare the alloy powders from the mechanical alloying. Therefore, the content is far from the title. The authors are suggested to revise the title or the content.

2. In the section of background, the authors are suggested to give more introduction on the development of Fe-Cr-Si alloy, especially its application.

3. In the experimental or results, the authors are suggested to give more characterization on the initial powders of Fe, Cr and Si. The morphology, size and crystal state are all helpful to the understanding of the following content.

4. In the content, the authors are suggested to calibrate diffraction peaks with phase or detailed crystallographic plane.

5. In the content, the authors provide the SEM observations on the powders with different content. Clearly, the chemical composition, mechanical milling time and heat treatment would influence the phase constituent and crystal state obviously. The authors could prepare the metallographic specimens of the powders and observe the microstructure of them, which could exhibit more information.

Author Response

Letter of response

We thank very much the referee for carefully reading the paper and for numerous comments. In all cases they were taken into account.

Referee 4

In the present research, the authors try to fabrication the Fe-Cr-Si alloy powders by the mechanical alloy method. The paper exhibits some results but there are some questions.

  1. The authors titled the paper as “Anti-Corrosion Properties of Fe-Cr-Si Alloys Prepared by Mechanical Alloying”. Based on the title, it seems the authors would fabricate the bulk Fe-Cr-Si alloy. Actually, they mainly prepare the alloy powders from the mechanical alloying. Therefore, the content is far from the title. The authors are suggested to revise the title or the content.

Discussion: The title has been corrected.

  1. In the section of background, the authors are suggested to give more introduction on the development of Fe-Cr-Si alloy, especially its application.

Discussion: The Introduction has been improved in the revised manuscript.

  1. In the experimental or results, the authors are suggested to give more characterization on the initial powders of Fe, Cr and Si. The morphology, size and crystal state are all helpful to the understanding of the following content.

Discussion: Initial powders of Fe, Cr and Si were obtained from chips and pieces powdered in planetary mill. This information has been added in the revised manuscript.

  1. In the content, the authors are suggested to calibrate diffraction peaks with phase or detailed crystallographic plane.

Discussion: Figures 1 and 2 have been corrected.

  1. In the content, the authors provide the SEM observations on the powders with different content. Clearly, the chemical composition, mechanical milling time and heat treatment would influence the phase constituent and crystal state obviously. The authors could prepare the metallographic specimens of the powders and observe the microstructure of them, which could exhibit more information.

Discussion: In the manuscript we provide a complex characterization of the prepared powders using several experimental techniques (XRD, SEM, TMS and XPS). We believe that these information are enough to properly describe the studied material.

Round 2

Reviewer 4 Report

The authors have revised the manuscript and answered the questions. Now, It is improved and could be accepted.